# Potential Role of the Circadian Clock in the Regulation of Cancer Stem Cells and Cancer Therapy

**DOI:** 10.3390/ijms232214181

**Published:** 2022-11-16

**Authors:** Yool Lee, Alfian Shan Tanggono

**Affiliations:** 1Department of Translational Medicine and Physiology, Elson S. Floyd College of Medicine, Washington State University, Spokane, WA 99202, USA; 2Steve Gleason Institute for Neuroscience, Washington State University, Spokane, WA 99202, USA

**Keywords:** circadian clock, cancer stem cells, epithelial–mesenchymal transition

## Abstract

Circadian rhythms, including sleep/wake cycles as well as hormonal, immune, metabolic, and cell proliferation rhythms, are fundamental biological processes driven by a cellular time-keeping system called the circadian clock. Disruptions in these rhythms due to genetic alterations or irregular lifestyles cause fundamental changes in physiology, from metabolism to cellular proliferation and differentiation, resulting in pathological consequences including cancer. Cancer cells are not uniform and static but exist as different subtypes with phenotypic and functional differences in the tumor microenvironment. At the top of the heterogeneous tumor cell hierarchy, cancer stem cells (CSCs), a self-renewing and multi-potent cancer cell type, are most responsible for tumor recurrence and metastasis, chemoresistance, and mortality. Phenotypically, CSCs are associated with the epithelial–mesenchymal transition (EMT), which confers cancer cells with increased motility and invasion ability that is characteristic of malignant and drug-resistant stem cells. Recently, emerging studies of different cancer types, such as glioblastoma, leukemia, prostate cancer, and breast cancer, suggest that the circadian clock plays an important role in the maintenance of CSC/EMT characteristics. In this review, we describe recent discoveries regarding how tumor intrinsic and extrinsic circadian clock-regulating factors affect CSC evolution, highlighting the possibility of developing novel chronotherapeutic strategies that could be used against CSCs to fight cancer.

## 1. Introduction

Circadian clocks are cell-autonomous timing systems that generate roughly 24 h periodic rhythms and are conserved in nearly all life, from unicellular organisms to humans. These internal timing systems integrate with diverse environmental (e.g., light, temperature) and metabolic (e.g., food intake) stimuli to regulate a myriad of temporal biological activities, including sleep/wake, feeding times, energy metabolism, hormonal and immune functions, and cellular proliferation and differentiation [1]. Evolutionarily, these flexible clock systems are thought to help organisms survive better by increasing their ability to timely anticipate and adapt to the cyclic changes of light and food availability as well as predation risk [2,3]. At the unicellular level, as in the case of all prokaryotes that are classified into bacteria and archaea, the timely compartmentalization of biochemical and redox processes by the intrinsic circadian clock ensures the temporal fitness of cell physiology and functions across species [4]. In lower eukaryotic species, including yeast and fungi, biological timekeepers may help these single-celled organisms escape the DNA-damaging effects of ionizing radiation from sunlight as well as oxidative stress during cell division [5,6,7]. In higher, complex organisms, such as most vertebrate species, including humans, multi-stimuli responsive oscillators in neurons and fibroblasts are tightly coupled to form the brain and peripheral organ clock systems, which, in turn, constitute a body-wide circadian network with adaptive capacity for the environmental entrainment of rhythmic behaviors and physiology [8].

In contrast to the evolutionarily adaptive benefits of circadian rhythms, rhythm disruptions, primarily due to modern lifestyles lacking regular patterns of working, eating, and sleeping, have been recognized to increase the susceptibility of humans to the onset and development of cardio-metabolic and immune disorders as well as multiple types of cancers [9,10,11,12]. Furthermore, several studies of animal models possessing genetic mutations in clock genes or exposed to forced circadian desynchrony regimens have strengthened the causal relationship between circadian disturbances and pathological consequences [13,14,15,16]. In particular, genetic and environmental perturbations of circadian rhythms largely alter the expressions and activities of several tumor suppressors and oncogenes in both host and tumor tissues in favor of cancer incidence and progression [17,18]. These circadian disruptions also reprogram host metabolism and immune systems, facilitating an immunosuppressive tumor microenvironment and enabling tumor progression and metastasis of multiple cancer types [13,14,19].

One of the most formidable issues in cancer diagnosis and treatment is cancer cell heterogeneity and plasticity, the adaptive capacity of diverse cancer cell populations to survive upon tumor-environmental challenges [20]. Cancer cells in tumor tissues are not uniform, but consist of distinct types of cell subpopulations that can change their cellular morphology, physiology, and behavior in response to tumor-intrinsic and extrinsic changes (e.g., gene mutations, low nutrient or oxygen levels, immune surveillance, or toxic drug exposure) [20,21]. Among the different tumor cell populations, the cancer stem cell (CSC), a self-renewing cancer cell type with multi-differentiation potential, is highly responsible for tumor recurrence and metastasis, therapy resistance, and mortality [21]. The generation of tumor cells with stem cell properties has been closely linked to the epithelial–mesenchymal transition (EMT), a process that increases the migratory and invasive capacities of cancer cells, leading to drug-resistance and metastatic potential in CSC populations [22,23]. Accordingly, effective eradication of these deadly cell populations during chemotherapeutic intervention is becoming more important for improved therapeutic outcomes. Growing evidence suggests that the cellular heterogeneity observed in several cancer types is not just genetically determined but also arises from inherent cancer cell plasticity instructed by the tumor-extrinsic microenvironment [24,25,26,27,28]. In line with this, recent cancer studies have shown that genetic (e.g., clock gene mutations) and environmental (e.g., chronic jet lag) factors that disrupt circadian rhythms highly impact the maintenance and/or properties of CSCs in tumors. In this review, we will discuss further details regarding the potential link between the circadian clock and tumor evolution, highlighting the possibility of developing treatment strategies that target CSC populations from the chronotherapeutic point of view.

## 2. The Molecular Clockwork and Circadian System

In all living organisms, the basic biological timing unit is the cell. In mammals, including humans, circadian rhythms are primarily driven by a cell-autonomous molecular feedback mechanism in which the circadian transcriptional activators aryl hydrocarbon receptor nuclear translocator-like (BMAL1) and circadian locomotor output cycles kaput (CLOCK) cyclically activate the expressions of their own repressors, period (PER) and cryptochrome (CRY) [29]. The core oscillator is complemented by a second loop in which cycling expression of BMAL1 is maintained by the REV-ERBα/β repressor and RORα/β activator proteins [29] (Figure 1A). Besides the core regulatory loops, multiple levels of epigenetic, posttranscriptional, and posttranslational regulation involving various kinases, phosphatases, ubiquitin-proteasome pathway components, nuclear-cytoplasmic transporters, non-coding RNAs, and chromatin remodelers contribute to the molecular clockwork, thus coordinating temporal programs via multiple clock-output genes involved in cellular physiology and metabolism [30,31,32,33,34,35,36,37]. The core clock machinery is mobilized to reset cellular rhythms upon intrinsic and extrinsic stimuli involving multiple post-translational (e.g., phosphorylation, sumoylation) and metabolic (e.g., Ca^2+^, cAMP) signaling pathways [32,36,38,39]. Systemically, oscillators that respond to multiple stimuli in neurons and fibroblasts are tightly coupled to form the brain and peripheral organ clock systems, which, in turn, constitute a body-wide circadian network with adaptive synchronous capacity for environmental entrainment of rhythmic behaviors (e.g., sleep/wake and feed/fast cycles) and physiology (e.g., temperature, blood pressure, energy metabolism, immune functions, cellular proliferation and differentiation) [30]. In particular, the hypothalamic suprachiasmatic nucleus (SCN), a central circadian pacemaker in the brain, communicates retinal light input to peripheral clock systems, thus mediating periodic synchronization of internal body rhythms with external day and night cycles [40] (Figure 1B). Besides SCN-mediated photic entrainment, a growing number of studies report that non-SCN brain regions (e.g., substantia nigra [SN], nucleus accumbens [NA], arcuate nucleus [ARC], choroid plexus [CR]), and peripheral tissues (e.g., liver, kidney, muscle, and adipose) possess their own autonomous and muti-stimuli entrainable oscillators that influence not only circadian functions in the SCN and neighboring local clocks, but also whole body rhythms via neural, hormonal, and metabolic feedback mechanisms [11,41,42,43,44,45,46,47,48,49].

## 3. Circadian Disruption and Cancer Pathogenesis

Mirroring the interactive features of biological clock systems, studies in both human and animal models have suggested that daily mismatches between the internal body clock and environmental cycles, such as irregular work (e.g., night shift) and life schedules (e.g., jet lag, mistimed eating), cause physiological complications that increase the risk of cardiac, metabolic, neurological, and neoplastic diseases [9,10,11,12]. Globally, artificial light sources in modern society are a primary contributor to disrupted circadian rhythmicity. Eighty percent of the world’s population is now exposed to light during the night, and about 18–20% of workers in the USA and Europe are engaged in night or rotating shift work, which predisposes them to multiple rhythm disorders, particularly cancer [50]. Indeed, several epidemiological studies suggest that night shift work and chronic jet lag increase the risks of onset and development of the most common cancer types (i.e., breast, lung, prostate, colorectal, and skin cancers) [51]. Based on such accumulating evidence, the International Agency for Research on Cancer (IARC) classified “shift-work that involves circadian disruption” as potentially “carcinogenic to humans (Group 2A)” in 2007 [52] and again in 2019 [53]. Besides work-related disruptions, additional factors that alter circadian rhythms, such as abnormal lighting conditions [54], irregular eating habits [55], and traumatic stress [56], may also contribute to tumor pathogenesis and progression. For example, a recent population-based case–control study reported that mistimed eating patterns increased the risks of breast and prostate cancers [57]. Overall, these findings underscore the importance of maintaining normal circadian rhythms for the prevention and management of cancer.

## 4. The Circadian Clock and Cancer Cell Proliferation

The relevance of the circadian clock to cancer pathophysiology has mainly been attributed to the regulatory relationship between the circadian clock and cell proliferation. Accumulating studies have unraveled molecular connections between the circadian clock and cell cycle-regulating factors [58,59]. Indeed, most of the genes regulating critical steps in the phases of the cell cycle have been found to be controlled by the circadian clock. For instance, c-MYC and cyclin D1, which induce entry into the DNA synthesis (S) phase, as well as WEE1, which controls entry into the mitosis (M) phase, exhibit circadian cycles of expression in a CLOCK/BMAL1-dependent manner [17,58,60,61,62]. There are also other cell cycle regulators and checkpoint controllers (e.g., p21, p16/INK4a, p27, and p57), as well as cell cycle signaling pathway factors (e.g., mitogen-activated protein kinase (MAPK), Wnt/β-catenin, and transforming growth factor (TGF)), that are regulated by the circadian clock to mediate daily cycles of cell proliferation [63]. Furthermore, the DNA damage response, DNA repair mechanisms, and apoptotic signaling pathways, which are closely associated with cell cycle checkpoints, are directly controlled by circadian clock components, such as CLOCK, PER1/2, and CRY1/2 in the core loop and RORA and NR1D1/2 in the stabilizing loop, of the molecular clockwork [64,65,66,67,68,69,70].

Notably, a growing number of studies have suggested direct roles of clock genes in tumor proliferation and growth. For example, genetic or epigenetic silencing of BMAL1 and/or CLOCK has been shown to increase tumor proliferation or growth rates in hematologic cancer [71], colon cancer [72], pancreatic cancer [73], tongue squamous cell carcinoma (TSCC) [74], breast cancer [75], lung adenocarcinoma [18], hepatocellular carcinoma (HCC) [76], nasopharyngeal carcinoma (NPC) [77], and glioblastoma multiforme (GBM) [78]. Conversely, overexpression of these clock activators suppresses the proliferative and malignant properties of tumor cells via cell cycle arrest and p53-dependent apoptosis [78,79]. In line with this, downregulation or upregulation of the BMAL1/CLOCK target genes PER1, PER2, and CRY1 promote or suppress tumor onset and proliferation, respectively, in Lewis lung carcinoma (LLC), mammary carcinoma [80], pancreatic cancer [81], lung carcinoma [18], leukemia [82,83], glioma [84], ovarian cancer [85], and oral squamous cell carcinoma [86,87]. At the molecular level, these studies suggest that the anti-cancer activity exerted by PER1/2 occurs via the inhibition of PI3K/AKT/mTOR-mediated glycolysis as well as cell cycle arrest and apoptosis induction [80,86,87]. In addition, recent studies from the Lamia group have reported that CRY1 and/or CRY2 promote the degradation of cMYC, early 2 factor (E2F) family members, and tousled like kinase 2 (TLK2) by recruiting these cell cycle-related oncogenic factors to the SCF^FBXL3^ ubiquitin-ligase complex [88]. Correlating with this, a recent large-scale systems analysis of 32 human cancer types has revealed that PERs and CRYs, along with several other clock genes, are downregulated in several types of cancers [89]. Taken together, these findings indicate that core clock components have tumor suppressor functions in most cancer types. However, most of these chrono-cancer studies have been based on bulk analyses of tumor cells grown in monolayer cultures or on tumor tissues grafted in mice, which are unable to recapitulate the spatiotemporal variations in phenotypes or the functions of distinct cancer cell populations in tumors under normal or abnormal circadian conditions [17].

## 5. The Circadian Clock and Cancer Stem Cell Heterogeneity and Plasticity

Stem cells are a dedicated group of cells with the potential to develop into many different types of cells in the body that are necessary for tissue differentiation, development, and homeostasis [90]. Recently, intensive stem cell research has demonstrated that the circadian clockwork plays an important role in the generation and maintenance of embryonic and adult stem cells during differentiation and tissue homeostasis (see these recent reviews for details [91,92]). In a growing number of studies, the genetic ablation of clock genes (e.g., *CLOCK*, *BMAL1*) in human and/or mouse stem cell populations has been shown to alter rhythmic oscillations in a variety of embryonic and adult stem cell-dependent processes, such as hematopoietic progenitor cell migration, the hair follicle cycle, bone remodeling, regenerative myogenesis, and neurogenesis [93]. Notably, similar to mouse embryonic stem cells (ESCs) and induced pluripotent stem cells (iPSCs) [94], human ESCs and iPSCs lack circadian rhythms, which was found to be primarily due to the post-transcriptional suppression of CLOCK protein expression [95,96]. In line with this, a recent study using human and/or mouse intestinal organoids and enteroids showed that the maturation of iPSC-derived intestinal organoids is required to develop robust circadian rhythms and gene expression that modulate circadian phase-dependent necrotic cell death responses to *Clostridium difficile toxin B* (TcdB), a toxin-producing intestinal pathogen [97]. These findings suggest that circadian clock functions not only control stem cell differentiation and development but also mediate time-of-day dependent responses during disease pathogenesis.

With the growing characterization of stem cell-specific markers (e.g., SOX2, OCT4, NANOG, CD133, CD44, ALDH1), as well as the development of lineage tracing (e.g., barcode technology, single-cell RNA sequencing) and three dimensional (3D) organoid technology in the area of stem cell research, it has become gradually clear that, such as normal tissues, cancer cells in tumor tissues are not homogenous but instead consist of diverse cell populations with different cellular lineages and properties [98]. Such intratumoral heterogeneity has been a major ongoing challenge in the effective therapeutic targeting of cancer. In particular, accumulating evidence suggests that CSCs (or cancer-initiating cells), which make up just a fraction of the cells within a tumor, are primarily responsible for the cancer cell diversity that results in therapeutic resistance and metastasis. Since the first discovery of malignant stem cell populations in tumors, a myriad of studies have reported on the identification of CSCs in many common cancer types, including leukemia [99,100,101], breast cancer [102], colorectal cancer [103,104,105], and brain cancer [106].

Interleukin-4 (IL-4) signaling, drug efflux proteins, and the upregulation of aldehyde dehydrogenase (ALDH) activity are all known to be associated with the resistance of CSCs to conventional cancer therapies [107]. The aberrant expression of genes within the Janus-activated kinase/signal transducer and activator of transcription (JAK/STAT), Hedgehog, Wnt, Notch, phosphatidylinositol 3-kinase/phosphatase and tensin homolog (PI3K/PTEN), and nuclear factor-κB (NF-κB) signaling pathways in various CSCs has also been reported to mediate drug/treatment resistance in these cells [104]. Phenotypically, CSCs have self-renewing capacity with a slow cycling rate relative to differentiated cancer cell populations [98]. This means that most anti-cancer drugs that target highly proliferative non-CSCs allow quiescent CSCs to escape, leading to tumor recurrence and therapy failure [22,98]. Thus, intensive pharmacological efforts have been made to screen for chemotherapeutic agents that selectively target CSC populations [108,109]. For example, a previous study using a chemical screening method discovered compounds (e.g., etoposide, salinomycin, and abamectin) that showed moderate-to-strong selective toxicity for breast CSCs [108]. One compound in particular, the potassium ionophore salinomycin, was found to reduce the proportion of CSCs by >100-fold relative to paclitaxel, a commonly used breast cancer chemotherapeutic drug, and inhibit mammary tumor growth in vivo [108].

Notably, a growing amount of evidence has shown that cancer cells in tumors exhibit an exceptional ability to dynamically switch between a stem cell state and a differentiated state [110] (Figure 2A). In several studies, such cancer cell plasticity has been suggested to be triggered by tumor intrinsic genetic variation and/or epigenetic modification, or systemic and external cues that affect the tumor microenvironment, such as hypoxia, cellular senescence, inflammation, injury, and drug exposure [109]. Furthermore, the trans-differentiation process of EMT has been identified as a key mechanism underlying cancer cell plasticity [111,112]. During EMT, cancer cells lose their round, cobblestone-like (epithelial) morphology and become elongated cells with fibroblast-like (mesenchymal) morphology, resulting in increased motility and invasion ability that is characteristic of malignant and stem cells [22] (Figure 2A). Indicative of the functional link between EMT and CSC phenotype, it has been shown that the tumor-initiating potential in several types of cancer cells, specifically breast cancer, melanoma, and glioblastoma cells, is enhanced by the overexpression of transcription factors involved in EMT activation, such as the zinc finger E-box binding homeobox genes ZEB1 and ZEB2 [111,112,113,114]. The loss of E-cadherin, a ‘caretaker’ of the epithelial phenotype, is also considered a key feature of EMT [115]. The E-cadherin/β-catenin complex plays a crucial role in maintaining the integrity of epithelial cell-to-cell contacts by keeping Wnt/ β-catenin signals in check [115]. Upon activation of Wnt signaling, disruption of the E-cadherin/-catenin complex leads to catenin release and subsequent translocation to the nucleus, where it promotes the transcription of EMT-associated genes (e.g., vimentin) [116]. EMT transition in CSCs has also been suggested to occur upon signals, such as growth factors and cytokines, released into the tumor-immune microenvironment by cancer-associated fibroblasts (CAFs) and tumor-associated macrophages (TAMs), or in response to hypoxia [110,117,118]. Due to the important relationship between EMT and cancer, several CSC-targeting anti-tumor agents have been generated to block or reverse EMT-related signaling pathways and/or gene expression to induce dedifferentiation of CSCs [111,112].

Accumulating evidence has suggested a potential role of the circadian clock in CSC heterogeneity and plasticity. Indeed, a recent study using monolayer and/or tumorsphere culture systems showed that C6 rat glioma and MCF-7 human breast tumor cells exhibited circadian rhythms in cellular morphology and population size related to CSC/EMT phenotype under stem cell-inducing media (SCM) conditions [119]. These in vitro results correlated with another earlier study showing that a number of stem-like cell populations with high ALDH1 activity in 4T1 breast cancer tumors implanted in mice exhibited circadian oscillations [120]. Circadian dynamics in CSC/EMT phenotypes may also explain, in part, a more recent study reporting that the metastatic spread of breast cancer cells accelerates during sleep both in humans and in mice, with the highest metastatic potential of circulating cancer cells (CTCs) being during the rest phase [121]. Notably, circadian disruptions using the chronic jet lag model have recently been shown to promote breast cancer cell dissemination and metastasis in mice by increasing the stemness and tumor-initiating potential of tumor cells and by creating an immunosuppressive shift in the tumor microenvironment [14]. In line with this, a more recent study using a rat model reported chronic circadian rhythm disturbance accelerated the proinflammatory responses that accompanied the activation of the canonical Wnt/β-catenin signaling pathway, which is closely linked to EMT in CSCs [122]. These results not only underscore the regulatory impacts of tumor intrinsic and/or extrinsic circadian functions in malignant tumor evolution but also indicate the importance of timed cancer therapy targeting circadian variations in the CSC/EMT phenotype (Figure 2B).

## 6. The Role of Circadian Clock Genes in Cancer Stem Cell Maintenance and Survival

### 6.1. The Role of CLOCK and BMAL1 in Cancer Stem Cells

In addition to the rhythmic properties of the CSC/EMT phenotype, emerging but limited studies have unraveled the functional roles of canonical circadian pathway components in CSC maintenance and survival (Figure 2C) (Table 1).

As noted above, circadian clock regulators have been suggested to function as tumor suppressors in most types of cancers. However, in recent loss-of-function studies using RNA interference (RNAi) and/or CRISPR genome editing technology, circadian molecular functions involving the core clock transcriptional activators CLOCK and/or BMAL1 have proven to be required for the maintenance of CSC/EMT properties in cancer cells [123,128,129]. In an earlier study using a murine model of acute myeloid leukemia (AML), disruption of *Clock* and/or *Bmal1* using RNAi in leukemic stem cells (LSCs) in vivo was shown to produce anti-leukemic effects, including impaired proliferation, enhanced myeloid differentiation, and depletion of LSCs [128]. Similar to these results, a subsequent GBM model study showed that downregulation of CLOCK or BMAL1 in glioblastoma stem cells (GSCs) induced cell-cycle arrest and apoptosis [123]. In further analysis, targeting CLOCK or BMAL1 was found to attenuate mitochondrial metabolic function and reduce the expression of tricarboxylic acid cycle enzymes, suggesting a critical role for these circadian activators in promoting stemness maintenance and metabolism in GSCs [123]. In addition to cancer cell stemness, the positive role of circadian activators in cancer malignancy was shown in a recent colorectal carcinoma (CRC) model. In this study, gene set enrichment analysis showed a significant enrichment of EMT and invasive signatures in BMAL1_high CRC patients compared to BMAL1_low CRC patients [126]. Correspondingly, *BMAL1* knockdown increased the expressions of epithelial markers (E-cadherin, CK-20, and EpCAM) but decreased the expressions of TWIST and mesenchymal markers (N-cadherin and vimentin) in CRC cell lines (HCT116 and SW480), promoting mesenchymal-epithelial transition (MET)-like changes in these cells [126]. As a result, BMAL1-depleted CRC cells exhibited decreased migration/invasion and drug resistance capacities, revealing the crucial role of BMAL1 in EMT-related CRC metastasis and chemoresistance [126].

In contrast, some studies using breast cancer cell models have reported opposite functions of circadian activators in CSC/EMT physiology. Studies of breast cancer stem cells (BCSCs) identified in aggressive breast tumors, such as triple-negative breast cancers (TNBCs), were found to exhibit high aldehyde dehydrogenase (ALDH) activity, which is associated with poor prognosis among breast cancer patients [102,130,131] and is considered to be a CSC marker for other types of cancers [132,133,134]. Furthermore, in a study from Dr. Shigehiro Ohdo’s lab using mouse 4T1 breast cancer cells, it was shown that the oscillation of clock gene expression was suppressed in ALDH-positive CSCs but not in surrounding cells in the tumor microenvironment [120]. In a subsequent study, among the clock genes, the expression levels of CLOCK mRNA and protein were found to be downregulated in ALDH-positive 4T1 cells. Moreover, enhanced expression of CLOCK was shown to exert a suppressive effect on the stemness and malignant properties of BCSCs, resulting in the prevention of tumor growth [125]. Similar to the role of CLOCK in suppressing stemness-related malignancy in breast cancer, recent transcriptomic and cell-based studies from Nuri Ozturk’s group have shown that BMAL1 disruption enhances EMT gene expression and the invasive properties of malignant breast cancer cells [75,135]. These results suggest divergent roles of CLOCK/BMAL1 in tumor evolution and pathogenesis, depending on the type and/or state of the cancer cell (Figure 2C).

### 6.2. The Role of PER2 and PER3 in Cancer Stem Cells

Unlike the mixed roles of CLOCK and BMAL1 in CSCs, period circadian regulator genes (e.g., PER2 and PER3), generally known as tumor suppressors in various types of cancer, have been shown to inhibit CSC maintenance and survival in tumors. For example, a recent study using a human GSC model showed that PER2 mRNA and protein expressions were downregulated in GSCs compared to non-stem glioma cells, suggesting that PER2 could be involved in the malignant process of glioma [124]. Moreover, PER2 overexpression was shown to induce the arrest of GSCs at the G0/G1 phase and suppresses their proliferation, stemness, and invasion ability in vitro and in vivo [124]. Mechanistically, PER2 suppresses the stem cell-like phenotype of GBM by dysregulating Wnt/β-catenin signaling, indicating that PER2 plays a critical role in regulating the stemness of GSCs [124]. Notably, these results are reminiscent of an earlier study showing that oncogenic transformation of PER2-defective cells resulted in the development of chemoresistance [136].

Similar to the role of PER2, another recent study using a prostate cancer stem cell (PCSC) model showed that PER3 (period circadian regulator 3) is a negative regulator of CSC biogenesis and phenotype. In this study, deep RNA-sequencing analysis using PCSCs (ALDH^hi^/CD44^+^) and non-PCSCs (ALDH^low^/CD44^–^) revealed that *PER3* is markedly downregulated in therapy-resistant PCSCs. Moreover, *PER3* overexpression in PCSCs was shown to significantly inhibit their clonogenicity and tumorigenicity, hence arresting tumor proliferation and development, while *PER3* knockdown in non-PCSCs was shown to dramatically promote their colony-forming and tumor-initiating abilities [127]. Mechanistically, low expression levels of PER3 result in the upregulation of BMAL1 expression, resulting in the phosphorylation of β-catenin and its translocation from the cytoplasm into the nucleus to activate the Wnt/β-catenin signaling pathway [127]. The prognostic relevance of PER3 function in cancer cell stemness is further indicated by a study showing that patients with high levels of PER3 expression have better survival rates than those with low levels of PER3 expression [127]. Collectively, these studies suggest that PER2 and PER3 are negative regulators of CSC physiology that could be potential targets for anti-CSC cancer therapy (Figure 2C).

## 7. Chronotherapeutic Approaches Targeting the Circadian Clock and CSCs

Given the functional roles of the molecular clockwork in cancer pathogenesis, it is becoming more important to take circadian mechanisms into account in the treatment of cancer [137,138]. In recent years, the idea of exploiting the circadian clock has been investigated in cancer therapy by modulating the activities of circadian clock molecules [129], enhancing intra-tumoral circadian rhythms [139], and optimizing anti-cancer drug delivery by timing it to the host’s circadian rhythms [140,141].

Notably, the presence of functional clocks in tumors is still debatable. It is commonly thought that circadian functions in cancer cells are largely compromised or deregulated, particularly due to mutations and/or the aberrant expression of oncogenes, such as Ras, p53, and Myc, that perturb the tumor clock [142,143]. However, some cancer cells, including U-2 OS human osteosarcoma and B16 mouse melanoma cells, exhibit marked circadian rhythms in a molecular clock-dependent manner, despite their highly tumorigenic and metastatic potentials [17,139,144,145]. Related to this, a more recent study using 2D and 3D mammalian culture models in vitro (mainly human U-2 OS cells) and ex vivo have shown that tumor cell oscillators can be coupled via paracrine pathways, particularly via TGF-β, which mediates phase adjustments of molecular clocks, similar to the intercellular coupling among cell-autonomous oscillators in normal tissues [146]. Furthermore, another relevant study showed that non-rhythmic MCF-7 human breast tumor cells, as well as C6 cells from rat glioma that have a well-established circadian clock, exhibit marked circadian rhythms in both *Per2* promoter activity and cellular morphology linked to CSC/EMT phenotype under stem cell media-based spheroid culture conditions [119]. In addition to experimental tumor models, patient-derived CSCs from glioblastoma or acute myeloid leukemia also display robust circadian rhythms that are similar to their differentiated cell counterparts [123,128]. These results point to the utility of exploiting intratumoral circadian rhythms in anti-cancer treatment.

In support of this notion, recent studies by Lee et al. have determined time-of-day specific activities of existing anti-tumor drugs, such as Palbociclib (PD0332991) [17], a Food and Drug Administration (FDA)-approved potent and selective CDK4/6 inhibitor, and 17-AAG [145], a heat shock protein 90 (HSP90) inhibitor, using human U-2 OS cell lines in vitro and in vivo mouse B16 melanoma tumor models. These time-specific drug actions were shown to occur in a tumor clock-dependent manner because genetic perturbation of *BMAL1* in the cancer cells via RNAi knockdown or CRISPR genome editing abrogated the temporal response of the tumor to the drugs [17,145]. Importantly, a more recent study using human TNBCs (MDA-MB-468LN, MDA-MB-231) reported that novel C-terminal heat shock protein 90 inhibitors (KU711 and KU758) as well as 17-AAG can target breast cancer stem cells and block migration, self-renewal, and epithelial–mesenchymal transition in tumorspheres in vitro and in in vivo tumor xenograft model experiments [147]. In line with this, the anti-tumor and anti-metastatic effects of the ALDH inhibitor N,N-diethylaminobenzaldehyde were shown to be enhanced by administration at the time of day when ALDH activity was high in 4T1 breast cancer stem cells that produce tumors in mice [120]. These findings suggest that existing or newly developed anti-cancer drugs can be repurposed to target CSC populations from chronotherapeutic perspectives.

Remarkably, recent studies have revealed that clock-targeting compounds, such as REV-ERB agonists (e.g., SR9009 and SR9011) and/or CRY protein stabilizers (e.g., KL001 and SHP656), exhibit potent anti-cancer properties in glioblastoma, leukemia, breast, colon, and melanoma cancer cell lines by inhibiting autophagy and de novo lipogenesis that is important for cancer cell survival [123,148,149]. In particular, some of these drugs (SR9009, SR9011, KL001) have been found to selectively inhibit patient-derived GSC proliferation and growth by suppressing the activation functions of CLOCK and BMAL1 in the maintenance of mitochondrial metabolic function and the expression of tricarboxylic acid (TCA) cycle enzymes [123]. Moreover, a recent in vitro study using human T98G cells, a useful GBM model, showed that dexamethasone-synchronized cells displayed differential, time-based responses to SR9009 treatment, with the highest responses being 18 to 30 h after synchronization [149]. This evidence suggests that chemotherapeutic use of clock targeting compounds can be combined with time-of-day specificity to increase their effectiveness.

## 8. Conclusions

CSCs, as a subpopulation of bulk tumors, have posed major challenges to conventional cancer treatments due to their drug-resistant and plastic properties involving EMT. So far, extensive pharmacological and chemotherapeutic efforts in both cancer research and clinical trials have been made to effectively target CSC/EMT pathways and improve treatment outcomes. In line with these efforts, the recent identification of the role of the circadian clock in CSC development presents an opportunity to find more effective therapeutic solutions by targeting tumor evolution from a chronobiological perspective. Unlike previous studies suggesting clock dysfunctions in tumors or tumor suppressive roles of clock genes, emerging evidence has revealed that circadian clocks function in favor of the development and maintenance of CSCs, particularly in brain and blood cancers. In addition to circadian gene oscillations, limited evidence has identified rhythmic properties in the cellular identities and compositions of CSC/EMT populations in tumors. However, the mechanisms underlying the circadian properties in CSC/EMT phenotypes are still debatable. Results from a recent 2D monolayer and 3D tumorsphere culture study using glioma and breast cancer cells suggested that daily changes in CSC/EMT phenotypes are cell-autonomous [119]. On the other hand, another metastatic breast cancer model study changes in the phenotypic rhythms of originally rhythm-negative CSCs can occur, primarily due to the paracrine effects of rhythm-positive non-CSCs via intercellular communications [125]. This discrepancy might be due to differences in the experimental methods and cell lines used and warrants further investigation, including research on other types of tumors. Regardless of either mechanism, the presence of circadian rhythms in CSC/EMT phenotypes may provide important insights and rationale for the development of chronotherapeutic approaches that target the time of day when the greatest number of invasive and metastatic cells are present in a tumor. In support of this notion, it was recently shown that circulating tumor cells (CTCs) that break away from breast tumors and metastasize to other parts of the body spread more aggressively during sleep in both humans and mice. Based on these results, it is tempting to assume that night-time delivery of anti-CSC/EMT agents is likely to be more helpful in improving treatment outcomes in patients with metastatic cancers. Furthermore, with the growing number of clock-targeting small molecules with CSC-specific toxicity being identified, it may be reasonable to expect that integrating chrono-cancer therapy into future research and clinical trials will improve the treatment and management of multiple cancers linked to CSC evolution.

## Figures and Tables

**Figure 1 ijms-23-14181-f001:**
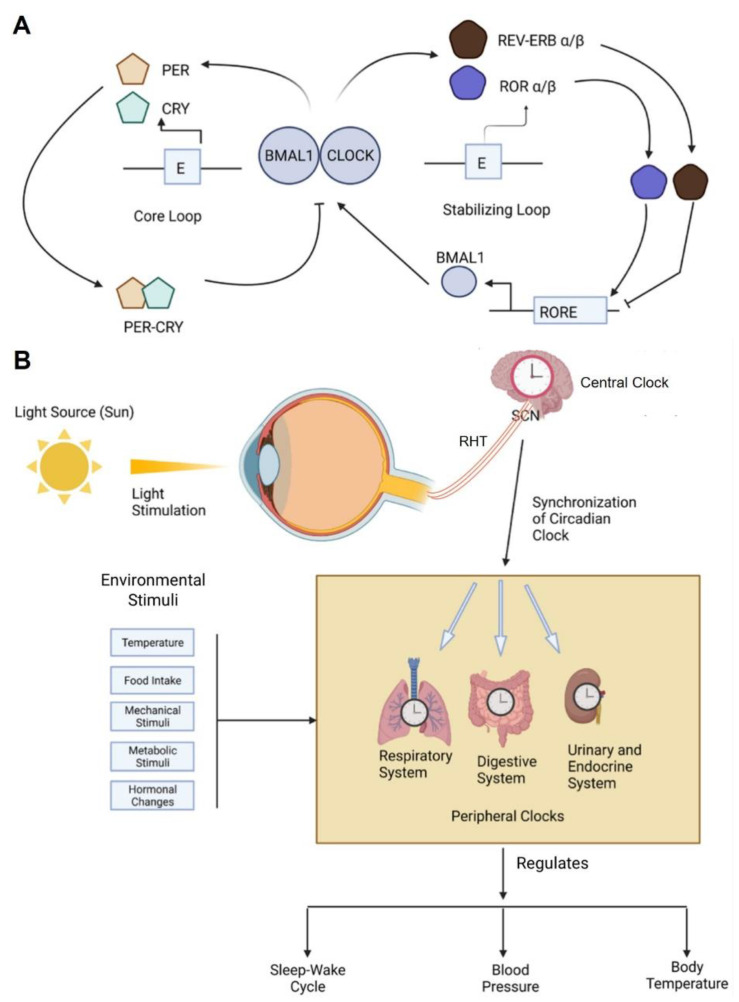
The core circadian timing systems in mammals. (**A**) Schematic of the molecular mechanisms of the circadian clock oscillator. At the molecular level, the circadian loop consists of a core loop and a stabilizing loop. In the core loop, the heterodimer BMAL1/CLOCK binds to the E box enhancer element (E) of the *PER* and *CRY* genes to initiate transcription. PER and CRY proteins accumulate before they heterodimerize and act as a transcriptional repressor. Expression of *BMAL1*, a core clock gene, is controlled by an ROR/REV-ERB-response element (RORE)-dependent mechanism, which plays an important role in stabilizing the period of the molecular circadian clock. (**B**) The suprachiasmatic nucleus (SCN), a small region in the brain, represents the highest hierarchy of core clock functions. When light stimuli are transmitted to the SCN through the retinohypothalamic tract (RHT), the SCN acts as a central pacemaker to synchronize the circadian clocks present in peripheral organs and coordinate temporal physiology. However, non-photic external cues (e.g., temperature changes, food intake, metabolic stimuli, hormonal changes) can reset circadian rhythms in peripheral clock tissues, thereby influencing the rhythmic output of physiology and behaviors.

**Figure 2 ijms-23-14181-f002:**
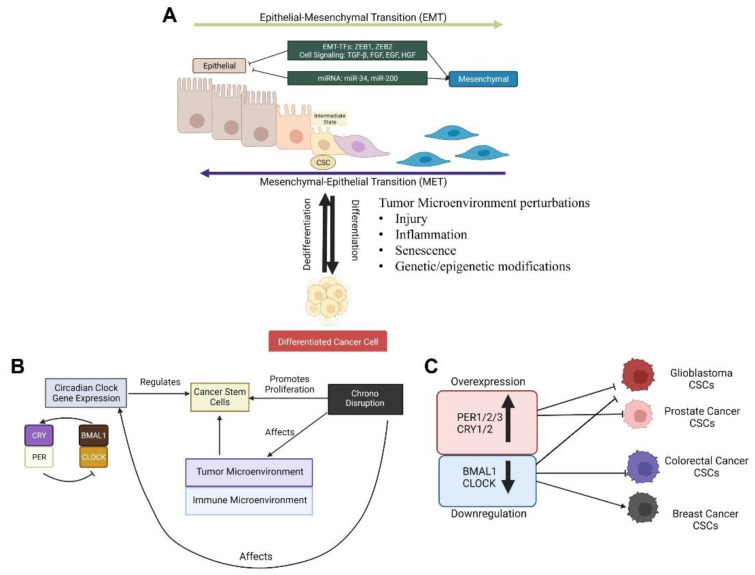
Role of the circadian clock in cancer stem cell plasticity and maintenance. (**A**) The generation and maintenance of cancer stem cells (CSCs) is associated with cellular plasticity in that the cells remodel themselves from a stem cell state (epithelial) into a differentiated state during a process called the epithelial–mesenchymal transition (EMT), according to tumor-microenvironmental changes. (**B**) Diagram showing the regulatory influences of molecular clock functions and circadian disruptions on CSC development. Circadian disruptions (e.g., chronic jet lag) remodel the tumor-immune microenvironment to favor the proliferation of CSCs. In turn, changes in the tumor microenvironment can alter circadian clock functions that contribute to the differentiation of cancer stem cells into malignant cancer cells. (**C**) Down-regulation of the BMAL1/CLOCK circadian activator complex or overexpression of the period genes (e.g., PER2 and PER3) inhibits CSC maintenance and survival in multiple types of cancers. One notable exception is a breast cancer stem cell model in which CLOCK repression was shown to increase tumor cell growth.

**Table 1 ijms-23-14181-t001:** The role of circadian clock genes in cancer stem cell phenotypes.

Cancer Stem Cell Type/Origin	Methods	Results	Ref.
Glioblastoma stem cells derived from patients undergoing surgical resection for glioblastomas at Duke University	BMAL1 and Clock Targeted shRNA knockdown; CRISPR/Cas9 Mediated Knockout of BMAL1 and CLOCK genes; Chip-Seq; RT-PCR; In Vivo Intracranial Tumor Formation	Knockdown of BMAL1 and CLOCK genes resulted in Cell Cycle Arrest and Apoptosis of Glioblastoma Stem Cells (GSCs); BMAL1 is repurposed to regulate tumor metabolism in GSC; BMAL1 and Clock contributes significantly to in vivo growth of GSC; Cell proliferation rate was substantially reduced in shBMAL1- and shCLOCK-transduced GSCs	Dong et al. [123]
Glioma Stem cells derived from U251 and U87 cell lines	Western Blot Analysis; RT-qPCR; In vivo Intracranial tumor formation; RNA-Seq; Immunofluorescence and immunohistochemical staining; Neurosphere formation assay; Cell proliferation assay and GSC flow cytometric assay	PER2 expression is downregulated in GSCs; PER2 overexpression could inhibit the stemness and self-renewal capability of GSCs; Overexpression of PER2 causes the G0/G1 cell cycle arrest, reducing proliferation rate; PER2 suppresses the stem cell-like phenotype of Glioblastoma by dysregulating Wnt/β-catenin signaling;	Ma et al. [124]
4T1 Breast Cancer tumors implanted in mice	Tumors plated under spheroid-forming condition; Transfection of plasmid vectors expressing products of circadian genes (BMAL1, CLOCK, CRY1, PER1, or PER2); 3D Culture chip; miRNA microarray Analysis; mIR-182 Knockout; RT-qPCR; Western Blot Analysis; Establishment of 4T1 cells stably expressing CLOCK; Invasion Assay; mRNA microarray analysis	CSC-like properties are observed in ALDH-positive cells; Expression levels of Clock mRNA were lower in ALDH-positive cells than ALDH-negative cells throughout the day; Attenuation of 4T1 cells invasive potential through EMT suppression by enhancement of CLOCK Expression; Suppression of tumor malignancy by enhancement of CLOCK expression in 4T1 cells; miR-182 suppresses the expression of Clock mRNA in ALDH-positive 4T1 cells; miR-182 Depletion suppresses growth in 4T1 cell-bearing mice	Ogino et al. [125]
Human colon cancer cell lines HCT116, SW480, and SW620 (CRC)	RNA-Seq Analysis; Western Blot Analysis; RT-qPCR; BMAL1 Gene Knockdown; Wound healing Assay; MTT Assay	BMAL1 knockdown Increases E-Cadherin Expression and E-Cadherin/β-Catenin Co-Localization at the Plasma Membrane of CRC Cells; BMAL1 Knockdown induces loss of mesenchymal traits and reinforce epithelial phenotypes; BMAL1 Knockdown decreases migratory capabilities and invasiveness of CRC cells	Zhang et al. [126]
PC3 and DU145 human prostate cancer cell line	Sphere and Colony-Formation Assays; Subcutaneous Injection of Cells into NOD/SCID Male Mice; RT-qPCR; PER3 Gene Overexpression and knockdown with lentiviral vectors; RNA-Seq; Purification of Double Marker Cells by Fluorescence-Activated Cell Sorting (FACS); Immunofluorescence; Western Blotting	Double Positive cancer cells exhibited higher sphere-forming capacity than Double Negative cells; PER3 Overexpression inhibits sphere-forming capabilities of Double Positive Cancer Cells, negatively regulates stemness in vitro; Per3 is downregulated in PC3 Double Positive Cells; PC3 Double Positive cells bear CSC features in vitro by expressing higher levels of CD44, ALDH1A1, and CD133 mRNA levels; PER3 negatively regulates stemness of PCSCs, probably via WNT/β-catenin signaling pathway	Li et al. [127]

## Data Availability

Not applicable.

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
