# Peer review of "Potential Role of the Circadian Clock in the Regulation of Cancer Stem Cells and Cancer Therapy"

_ijms, 2022, doi:10.3390/ijms232214181_

Round 1

Reviewer 1 Report

In this review, the authors described the potential role of the circadian clock in the regulation of cancer stem cell progression, highlighting the possibility of developing  novel potential chemotherapeutic strategies against CSCs. In particular, they well described and represented through figures the role of circadian clock in cancer stem cell plasticity and relationship among the molecular clock functions and circadian disruptions on CSC development. Furthermore, I found the area related the chemotherapeutic approaches very interesting, describing the chemotherapeutic use of clock targeting compounds, in order to optimize the anti-tumor and anti-metastatic effect. Finally, the references used in this review are adequate and quite informative for the readers. In conclusion, I consider this review well organized, with a high contribution to the research field described and potentially useful for future research works. I have no further suggestions for the authors. In my opinion the paper is ready to be published in this form.

Author Response

We are grateful for the generally positive remarks from the reviewers and for the feedback provided by them. We have addressed concerns as follows:

Reviewer 1

In this review, the authors described the potential role of the circadian clock in the regulation of cancer stem cell progression, highlighting the possibility of developing  novel potential chemotherapeutic strategies against CSCs. In particular, they well described and represented through figures the role of circadian clock in cancer stem cell plasticity and relationship among the molecular clock functions and circadian disruptions on CSC development. Furthermore, I found the area related the chemotherapeutic approaches very interesting, describing the chemotherapeutic use of clock targeting compounds, in order to optimize the anti-tumor and anti-metastatic effect. Finally, the references used in this review are adequate and quite informative for the readers. In conclusion, I consider this review well organized, with a high contribution to the research field described and potentially useful for future research works. I have no further suggestions for the authors. In my opinion the paper is ready to be published in this form.

Thank you very much for the positive comments.

Reviewer 2

I consider this paper suitable to be published after minor revision:

1.The title of Figure 1 need to be bold - like the title of Figure 2

Thank you for the catch. We corrected it (Page 3, Line 115).

  1. Table 1 need to be titled.

Thank you for the suggestion. We added the title for Table 1 (Page 8, Line 313).

  1. The literature need to be carefully checked out and corrected; for instance, in the lines 784 to 786 and in the lines 787 to 789 the title of the source, its volume and pages are lacking; furthermore, in the lines 790 to 792 and 823 there is no volume and pages for the sources.

Thank you for the important note. We checked and corrected the references with volume and page info.

Reviewer 2 Report

I consider this paper suitable to be published after minor revision:

1. The title of Figure 1 need to be bold - like the title of Figure 2.

2. Table 1 need to be titled.

3.  The literature need to be carefully checked out and corrected; for instance, in the lines 784 to 786 and in the lines 787 to 789 the title of the source, its volume and pages are lacking; furthermore, in the lines 790 to 792 and 823 there is no volume and pages for the sources. 

Author Response

We are grateful for the generally positive remarks. We have addressed concerns as follows:

I consider this paper suitable to be published after minor revision:

1.The title of Figure 1 need to be bold - like the title of Figure 2

Thank you for the catch. We revised it (Page 3, Line 115).

  1. Table 1 need to be titled.

Thank you for the suggestion. We added the title for Table 1 (Page 8, Line 313).

  1. The literature need to be carefully checked out and corrected; for instance, in the lines 784 to 786 and in the lines 787 to 789 the title of the source, its volume and pages are lacking; furthermore, in the lines 790 to 792 and 823 there is no volume and pages for the sources.

Thank you for the important note. We checked and corrected the references with volume and page info.